# Tenuazonic Acid-Triggered Cell Death Is the Essential Prerequisite for *Alternaria alternata* (Fr.) Keissler to Infect Successfully Host *Ageratina adenophora*

**DOI:** 10.3390/cells10051010

**Published:** 2021-04-25

**Authors:** Jiale Shi, Min Zhang, Liwen Gao, Qian Yang, Hazem M. Kalaji, Sheng Qiang, Reto Jörg Strasser, Shiguo Chen

**Affiliations:** 1Weed Research Laboratory, Nanjing Agricultural University, Nanjing 210095, China; 2019216009@njau.edu.cn (J.S.); zhangmin0610@163.com (M.Z.); 2019116002@njau.edu.cn (L.G.); 2020116004@stu.njau.edu.cn (Q.Y.); wrl@njau.edu.cn (S.Q.); strasserreto@bluewin.ch (R.J.S.); 2Department of Plant Physiology, Institute of Biology, Warsaw University of Life Sciences SGGW, 159 Nowoursynowska 159, 02-776 Warsaw, Poland; hazem@kalaji.pl; 3Institute of Technology and Life Sciences, Falenty, Al. Hrabska 3, 05-090 Raszyn, Poland; 4Bioenergetics Laboratory, University of Geneva, Jussy, CH-1254 Geneva, Switzerland

**Keywords:** necrotrophic pathogen, mycotoxin, reactive oxygen species, disease susceptibility

## Abstract

The necrotrophic fungus *Alternaria alternata* contains different pathotypes that produce different mycotoxins. The pathotype *Ageratina adenophora* secretes the non-host-selective toxin tenuazonic acid (TeA), which can cause necrosis in many plants. Although TeA is thought to be a central virulence factor of the *A. adenophora* pathotype, the precise role of TeA in different stages of host infection by pathogens remains unclear. Here, an *A. alternata* wild-type and the toxin-deficient mutant *ΔHP001* with a 75% reduction in TeA production were used. It was observed that wild-type pathogens could induce the reactive oxygen species (ROS) bursts in host leaves and killed photosynthetic cells before invading hyphae. The ROS interceptor catalase remarkably inhibited hyphal penetration and invasive hyphal growth and expansion in infected leaves and suppressed necrotic leaf lesion. This suggests that the production of ROS is critical for pathogen invasion and proliferation and disease symptom formation during infection. It was found that the mutant pathogens did not cause the formation of ROS and cell death in host leaves, showing an almost complete loss of disease susceptibility. In addition, the lack of TeA resulted in a significant reduction in the ability of the pathogen to penetrate invasive hyphal growth and spread. The addition of exogenous TeA, AAL-toxin, and bentazone to the mutant *ΔHP001* pathogens during inoculation resulted in a significant restoration of pathogenicity by increasing the level of cell death, frequency of hyphal penetration, and extent of invasive hyphal spread. Our results suggest that cell death triggered by TeA is the essential requirement for successful colonization and disease development in host leaves during infection with *A. adenophora* pathogens.

## 1. Introduction

Generally, fungi use three different nutritional strategies, including necrotrophy, hemibiotrophy, and biotrophy, to infect plants [1]. Necrotrophic fungi exert their destructive effects on plants by producing cell wall-degrading enzymes and proteins or secreting low molecular weight secondary metabolite toxins, which often play a key role in controlling pathogenicity or virulence [2,3,4]. Such fungal toxins alone can reproduce some or even all of the symptoms of infection caused by the living pathogen organisms [5]. In general, necrotrophic fungal pathogens use toxins as pathogenicity or virulence factors to induce cell death of invaded plants and to obtain food for their growth and colonization from the dead tissue [3,6].

The fungal genus *Alternaria* is a highly diverse group of necrotrophic plant pathogens due to its broad host range and worldwide distribution. *Alternaria* species cause diseases in more than 400 plant species, including many important crops and weeds, resulting in severe economic losses in agricultural production. Among them, *A. alternata* is the most common and can infect over 100 plant species alone [7,8]. All plant pathogenic *Alternaria* species can produce different host selective toxins (HSTs) or non-host selective toxins (NHSTs), both of which are essential for disease development in plants and successful completion of the pathogenic life cycle. This could be an important reason for the success of these *Alternaria* pathogens [8]. *A. alternata* (Fr.) Keissler has several pathogenic variants, each producing a unique toxin and causing disease in different plants. To date, over 70 toxins isolated from *Alternaria* species have been reported [9,10,11]. The major *A. alternata* HSTs with identified chemical structures are the AAL-toxin from tomato pathotype, AK-toxin from Japanese pear pathotype, ACR-toxin from Rough lemon pathotype, AM-toxin from apple pathotype, AF-toxin from strawberry pathotype, AT-toxin from tobacco pathotype, and ACT-toxin from mandarin pathotype [9,12,13]. In addition to HSTs, many *Alternaria* NHSTs have also been identified, such as tenuazonic acid (TeA), tentoxin and brefeldin A, alternaric acid, and zinniol [9,12]. These toxins may be involved in some or all stages of infection of *Alternaria* pathogens by interfering with various biochemical processes and inducing the formation of reactive oxygen species (ROS) and cell death [11,13]. The role of NHSTs of *A. alternata* has been extensively studied, which are released from germinating conidia prior to penetration aid and are the primary virulence factors [13]. NHSTs may not be the primary determinant of disease, but may serve as important virulence factors in one or more periods of infection. Often their role in plant diseases is too complex to be clearly determined [2,7].

Qiang [14] isolated a new *A. alternata* pathogen from diseased leaves of an invasive plant *Ageratina adenophora*, which was identified as an *A. adenophora* pathotype. It was concluded that the secondary metabolites should be responsible for the pathogenicity of the *A. adenophora* pathotype [15]. Based on extensive experiments on compound isolation, purification, and structure identification, the main active metabolite was found to be TeA [16]. As a known *A. alternata* toxin, TeA was first isolated from culture filtrates of *A. tenui* in 1957 [17]. However, most studies on TeA for a long time focused only on its sources, toxicity to animals, and pharmaceutical activities [18]. Our previous studies have shown that TeA is a novel photosynthetic inhibitor and has the potential to be developed as a bioherbicide due to its broad-spectrum herbicidal activity [19,20,21]. TeA can cause chloroplast-derived ROS burst by inhibiting electron transport on the acceptor side of photosystem II (PSII) and chloroplast ATPase activity, resulting in plant cell death and tissue necrosis [22]. Further studies suggested that TeA-induced ^1^ O_2_ activates the EXECUTER -protein-dependent signaling pathway and triggers a cell death response in *Arabidopsis* seedlings [23]. Recently, Kang et al. [24] found that the TeA-deficient mutant *ΔHP001* of *A. alternata* lost its pathogenicity on the host *A. adenophora*, but the addition of TeA partially restored its pathogenicity. The hyphae of the *ΔHP001* mutant did not form appressoria, but some appressorial structures were observed when TeA was added on the onion epidermis. Moreover, it was found that TeA is of high importance for maintaining mycelial development and morphology on the medium. Thus, it can be concluded that TeA is an important virulence factor of *A. adenophora* pathotype. However, the exact mechanism of TeA at different stages of host infection by *A. alternata* remains unclear.

In this work, we hypothesized that cell death triggered by TeA is the essential requirement for hyphal penetration, invasive hyphal growth and spread, and disease development when the host *A. adenophora* leaves were infected with *A. alternata*. To test this hypothesis, the toxin-deficient mutant strain *ΔHP001* was used as a good genetic material to investigate the role of TeA in the different stages of infection by *A. alternata* in the host. Moreover, we tried to propose a model for the role of TeA in *A. alternata* infection in the host plant.

## 2. Materials and Methods

### 2.1. Plant Materials and Chemicals

*A. adenophora* plants were grown in a mixture of soil and peat (1:1, *v*/*v*) at 20–25 °C under 300 μmol m^−2^ s^−1^ white light (day/night, 12/12 h) and relative humidity (≥60%) in a greenhouse. The second and third top mature leaves of the healthy plants were used to perform further experiments.

TeA, AAL-toxin, bentazone, trypan blue, 3,3′-diaminobenzidine (DAB), nitroblue tetrazolium (NBT), catalase (CAT), methanol, and acetonitrile were purchased from Sigma-Aldrich (Shanghai, China), and other common chemical reagents used in this study were obtained from Amresco (Solon Ind PkwySolon, OH, USA). TeA, AAL-toxin, and bentazone stock solutions were dissolved in 100% methanol and further diluted with sterile water. The final concentration of methanol used in all chemicals was less than 0.1% (*v*/*v*).

### 2.2. Fungal Strains and Culture Conditions

The *Alternaria alternata* (Fr.) Keissler wild-type strain NEW001 (WT) isolated from an invasive plant *Ageratina adenophora* [15] and toxin-deficient mutant *ΔHP001* (MU, *HP001* gene knockout strain) [24] were used in this work. They were cultivated at 25 °C on 40 mL of potato dextrose agar (PDA) medium in a 9 cm (diameter) glass petri dish. Collection of mycelia was as described [24]. Ten 3-mm-diameter agar discs taken from the margin of a 3-day-old colony of wild-type or mutant strains grown on PDA medium were transferred into a new glass flask containing 150 mL of potato dextrose broth (PDB) medium (potato 200 g L^−1^, sucrose 20 g L^−1^) and grown at 28 °C, 110 rpm for 4 days under dark conditions. Mycelium cultural solutions were centrifuged at 1000× *g* (Allegra^™^ 64R centrifuge, Beckman, Coulter, CA, USA) for 5 min and the supernatants were removed to get the deposited mycelia. For preparing mycelial homogenates, 500 mg deposited mycelia were taken and mixed with sterile water to get a 1.5 mL suspension, and then smashed using a homogenizer. For collection of conidia, *A. alternata* wild-type strains were grown on PDA medium for 15 days. *A. alternata* mutant *ΔHP001* strains were grown on PDA medium for 7 days, subsequently, transferred to a black light condition for 2 days. Conidia were harvested with sterile water and filtered using filter paper. Conidia were resuspended in sterile water with 0.02% Tween-20, the concentration was adjusted to 8 × 10^5^ conidia mL^−1^ before inoculation.

### 2.3. Pathogen Inoculation

Detached *A. adenophora* leaves were inoculated with 20-μL droplets of mycelial homogenates of *A. alternata* wild-type and mutant *ΔHP001*. In experiments of application of TeA, AAL-toxin, and bentazone, 10 μL droplets of sterile water, 20 μg mL^−1^ TeA, 20 μg mL^−1^ AAL-toxin or 200 μg mL^−1^ bentazone were deposited on the surface of leaves. After the leaves were incubated for 1 h at 25 °C in darkness, 20-μL droplets of mycelial homogenates of *A. alternata* mutant *ΔHP001* were inoculated on the pretreated site. For effects of ROS scavenger, before inoculation of *A. alternata* mycelial homogenates leaves were pretreated for 1 h with 5-μL droplets of sterile water or 300 U mL^−1^ catalase in the dark. In conidia inoculation, detached leaves were immerged into a conidia suspension (8 × 10^5^ conidia mL^−1^) for 3 min. In addition, to observe conidia germination and development of *A. alternata* mutant *ΔHP001*, detached leaves were also sprayed with 20 μg mL^−1^ TeA, 20 μg mL^−1^ AAL-toxin or 200 μg bentazone and then incubated for 1 h at 25 °C in darkness prior to conidia inoculation. All of the inoculated leaves were placed in petri dishes lined with moistened (sterile water) filter paper and kept at 25 °C and high relative humidity (>90%) in a growth chamber (E-36HO, Percival Scientific Inc., Perry, IA, USA).

### 2.4. Measurement of TeA Content

TeA extraction and quantification were referred to Oviedo et al. [25] with some modifications. Briefly, after 4 days of *A. alternata* wild-type or mutant *ΔHP001* strains grown in PDB liquid medium, mycelia were harvested by centrifugation at 1000× *g* for 5 min. After the supernatants were removed, 100 mg fresh deposited mycelia were used to extract TeA in 4.5 mL chloroform/methanol (2:1, *v*/*v*) for 60 min in an ultrasonic bath. The extract was transferred to a clean 50 mL centrifuge tube. The same mycelia were then extracted ultrasonically for 60 min in 3.9 mL ethyl acetate containing 1% formic acid. The second extract was transferred to the centrifuge tube containing the first dried extract. The mycelia were then extracted ultrasonically for 60 min with 4.5 mL of isopropanol and the extract was transferred to the centrifuge tube with the two previous extracts. The total extracts were dried under a vacuum with a lyophilizer (LGJ-10, Xinyi Inc., Beijing, China) and were re-dissolved ultrasonically in 5 mL of 80% methanol, and then filtered through a GF/B 0.22-μm glass microfiber filter (Whatman, Kent, UK) and transferred to a clean 1.5 mL amber vial for high-performance liquid chromatograph (HPLC). Chromatograph separation was performed on the Agilent Infinity 1260 HPLC instrument (Agilent Technologies Inc., Santa Clara, CA, USA) using an Agilent ZORBAX SB-Aq Stable Bond Analytical column (250 × 4.6 mm, 5 μm) (Agilent Technologies Inc., Santa Clara, CA, USA). The detector was set to 280 nm. The mobile phase is a methanol/water mixture solution (80:20, *v*/*v*) and a flow rate of 1.0 mL min^−1^ was carried out. The injection volume was 50 μL and the retention time of TeA was 2.7 min. Quantification was relative to external standards of 1.0, 2.5, 5.0, and 10 μg/mL in 80% methanol. The analytical standard tenuazonic acid (37018-0.1MG) was purchased from Sigma-Aldrich (Shanghai, China).

### 2.5. Pathogenicity and Cell Death Assays

Pathogenicity after inoculation was evaluated by determining the necrotic lesion diameter of at least 15 leaves per treatment. To visualize cell death of leaf tissues, trypan blue staining was used as described in Joo et al. [26]. Detached-leaves after inoculation were immerged into a trypan blue mixture (30 mL ethanol, 10 g phenol, 10 mL H_2_O, 10 mL glycerol, 10 mL of 10.8 M lactic acid, and 10 mg of trypan blue) and boiled for 3 min in a water bath. The leaves were left overnight in the staining solution. The next day, the leaves were transferred to a chloral hydrate distaining solution (2.5 g mL^−1^, pH 1.2) and boiled for 10 min, and left at room temperature for at least 24 h. The decolorized leaves were recorded using a digital camera (G15, Canon Inc., Ōta, Tokyo Japan).

### 2.6. Chlorophyll a Fluorescence Imaging

To monitor the early effect of *A. alternata* infection on leaf photosynthesis, chlorophyll fluorescence imaging was applied using a pulse-modulated Imaging-PAM M-series fluorometer (MAXI-version, Heinz Walz GmbH, Effeltrich, Germany). Leaves inoculated with mycelial homogenates were placed under the imaging system camera for 30 min for dark-adaptation. At a weak measuring light (0.25 µmol m^−2^ s^−1^) and saturation pulse (6000 µmol m^−2^ s^−1^), the minimum (F_O_) and the maximum (F_M_) fluorescence yield were determined respectively, from which the value of maximum quantum efficiency of PSII (F_V_/F_M_) was calculated.

### 2.7. Scanning Electron Microscopy (SEM)

The SEM observation was according to Becker et al. [27] with minor modifications. After inoculation of *A. alternata* conidia for 3 to 48 h, three leaf segments (0.5–1.0 cm) were prepared with fresh razor blades from each inoculated leaf and fixed under a vacuum in 2.5% (*v*/*v*) glutaraldehyde in 0.1 M phosphate buffer (pH 7.2) for 2 h at room temperature and stored overnight at 4 °C in the same fixative buffer. Leaf samples were washed three times with 0.1 M phosphate buffer (pH 7.2) and dehydrated by passing through a graded ethanol series (70%, 80%, 90%, and 100%) for a period of 30 min in each gradient. The samples were dried at a critical point with liquid CO_2_. The fixed materials were coated with a 10-nm layer of gold/palladium (60:40), and observed using a HITACHI S-3000N scanning electron microscope (Hitachi, Tokyo, Japan).

### 2.8. Light Microscopy

For examination of hyphal penetration and growth on leaves, leaf segments from a specified region (circular area with a radius of 1 to 4 mm from the primary inoculation site outwards) were collected after leaves were inoculated with mycelial homogenates. Lactophenol-trypan blue staining (10 mL lactic acid, 10 mL glycerol, 10 g phenol, 10 mg trypan blue, 10 mL sterile water) was used to visualize hyphae as previously described [28]. Samples were viewed and recorded using a light microscope (Carl Zeiss Axio Imager M2; Carl Zeiss Microscopy GmbH, Oberkochen, Germany).

### 2.9. ROS Burst Detection

The accumulation of hydrogen peroxide (H_2_O_2_) and superoxide radicals (O_2_^•−^) was determined using DAB and NBT as the substrate [22]. Detached leaves inoculated with mycelial homogenates at 3, 6, and 9 h post inoculation were vacuum infiltrated with either 1 mg mL^−1^ DAB-HCl (pH 3.8) or 0.1 mg mL^−1^ NBT in 25 mM HEPES-KOH buffer (pH 7.6) and incubated for 4 h at room temperature in the dark. Chlorophyll was removed by immersing the leaves in boiling ethanol (95%) for 10 min. Afterwards, the leaves were stored in 95% ethanol and recorded with a digital camera (COOLPIX 4500, Nikon Inc., Tokyo, Japan). H_2_O_2_ and O_2_^•−^ were visualized as a reddish-brown and dark-blue coloration.

### 2.10. Statistical Analysis

One-way ANOVA was carried out and means were separated by Duncan LSD at 95% using SPSS Statistics 20.0.

## 3. Results and Discussion

### 3.1. TeA Deficiency Decreased the Pathogenicity of A. alternata in A. adenophora Leaves

To evaluate the importance of TeA in the pathogenicity of *A. alternata*, the TeA content of mycelia harvested from a PDB liquid medium was measured. As shown in Figure 1a, the TeA content in the toxin-deficient mutant *ΔHP001* is three times lower than in the wild-type. The yield of TeA may be regulated by the Fus3/Kss1 MAPK pathway of *A. alternata* [24]. The TeA content of 20-μL droplets of mycelial homogenates was calculated to be approximately 0.1 μg higher in the wild-type than in the *ΔHP001* mutant. Therefore, 0.2 μg TeA, which is twice the variation, was used to investigate the effect of exogenous TeA on *A. alternata* infection.

After inoculation of mycelial homogenates on *A. adenophora* leaves, the mutant strain *ΔHP001* showed a significantly slower growth rate than the wild-type. Large amounts of grey aerial hyphae were formed as early as 24 h after inoculation (hpi) compared to the wild-type mycelium. However, 48 h after inoculation of the mutant mycelia, very few aerial hyphae are present. The application of exogenous TeA was able to stimulate hyphal growth and production of aerial hyphae on mutant-inoculated leaves (Figure 1b). Kang et al. [24] found that TeA efficiency in the *ΔHP001* mutant significantly affected colony morphology, aerial hyphae growth, and mycelial cell wall thickness, but the addition of exogenous TeA partially restored these influences. These changes in the biological properties of the *ΔHP001* mutant due to the TeA loss may account for its slow growth rate and low pathogenicity on host leaves.

After 48 h of inoculation of the *ΔHP001* mutant, no visible disease lesion appeared on leaves, but a necrotic lesion was formed at 24 hpi and developed rapidly at 48 hpi on wild-type inoculated leaves (Figure 1b). This indicates that the TeA-deficient mutant *ΔHP001* lost its pathogenicity. Small necrotic lesions were observed at 48 hpi of the *ΔHP001* mutant with exogenous TeA (Figure 1e), indicating that exogenous TeA could partially restore the pathogenicity of the mutant. These results support that TeA is one of the most important virulence factors of *A. alternata* [24]. Many mycotoxins produced by fungi are pathogenicity or virulence factors and play an important role in initiating or exacerbating plant diseases [29]. For several plant pathotoxins of *A. alternata*, toxins such as AK-toxin [30], AM toxin [31,32], and AAL-toxin [33] have been shown to be essential for their pathogenicity. Dangl and Jones [34] suggested that toxins secreted by necrotrophs induce cell death, leading to the formation or expansion of necrotic lesions in infected plant tissues. Here, cell death was detected by trypan blue staining. As a result, cell death occurred at 12 hpi on leaves and a progressive range of death was observed in the case of wild-type infection. The *ΔHP001* mutant did not cause obvious cell death on leaves at 48 hpi unless exogenous TeA was added (Figure 1b). Obviously, the appearance of cell death is an earlier event than the appearance of the disease lesion on leaves inoculated with the wild-type. It is concluded that TeA is required for the pathogen of *A. alternata* to induce cell death to promote disease development.

### 3.2. TeA Played a Crucial Role in A. alternata Infection Process of A. adenophora Leaves

Parasitic plant fungi form infection structures that complete crucial stages of pathogenesis, including attachment, host recognition, penetration, proliferation, and feeding [35]. To investigate the role of TeA in different stages of *A. alternata* infection, *A. adenophora* leaves were evaluated by scanning electron microscopy after inoculation with a conidial suspension (8 × 10^5^ conidia mL^−1^) of the wild-type or the *ΔHP001* mutant (SEM). SEM observations showed that there was no difference in conidial germination, germ tube, and mycelial development between the wild-type and mutant in the early phase of infection (Figure 2). Their conidia started to germinate as early as 3 hpi, and then the mycelia spread by unhindered growth on the leaf surface.

Gudesblat et al. [36] showed that stomata are a major route into plant tissues for many plant pathogens. Here, penetration of hyphae into the leaf tissue via stomata (Figure 3b) or directly through epidermal cell junctions or cracks (Figure 3a,c) was observed from SEM 24 h after inoculation of *A. alternata* wild-type. Interestingly, stomata appeared to be a preferential site for hyphae penetration into leaf tissues during the early phase of infection after inoculation of wild-type conidia. At 36 and 48 hpi, deep cracks were formed due to the severe destruction of the cuticle at the site of contact directly with the hyphae of *A. alternata* wild-type, which penetrated the leaf tissue mainly through these cracks and occasionally through the stomata (Figure 3d–f). Thus, many subcuticular hyphae were observed along the entire length of the cuticular crack. Pochon et al. [37] also found that penetration of *A. brassicicola* hyphae into *Arabidopsis* seeds occurs mainly through cracks in the seed coat. Our results differ from previous observations that penetration of *A. alternata* hyphae into tissues after conidial inoculation occurred only through the stomata [14]. In contrast, after 36 h of inoculation, the conidia of the *ΔHP001* mutant still failed to penetrate the tissue but instead continued to grow on the outer surface of the leaf epidermis (Figure 3g). At this stage, the mutants also failed to break through the cuticle and then form cracks penetrating into the hyphae. This suggests that TeA deficiency suppresses the leaf cuticle destruction caused by *A. alternata*. When leaves of the *ΔHP001* mutant were inoculated with exogenous TeA for 36 h, penetration of some hyphae into the tissue over the stomata was observed (Figure 3i,j). It was suggested that TeA plays a crucial role in the penetration of *A. alternata* hyphae into the leaf tissue. Remarkably, a few hyphae of the *ΔHP001* mutant also caused rupture of the leaf cuticle and formed cracks through which the hyphae of the mutant conidia penetrated into the tissue at 48 hpi (Figure 3h). Obviously, the *ΔHP001* mutant showed a much weaker ability of hyphal penetration into the leaf tissue compared with the wild-type. This phenomenon can be attributed to the fact that the toxin-deficient mutant *ΔHP001* does not completely lose the ability to produce TeA and can still secrete some TeA (Figure 1a). The low TeA content in the *ΔHP001* mutant drastically reduced the ability of hyphae to invade the leaf tissue.

To further confirm the role of TeA in the early stages of *A. alternata* infection, invasion and invasive hyphal development in leaf tissue were visualized by staining the hyphae with lactophenol trypan blue and observing the tissue under a light microscope. An experimental model of mycelial inoculation and sampling was designed to observe the growth and expansion of wild-type and mutant *ΔHP001* mycelia (Figure 4a). At 3 hpi, the mycelial pieces of the wild-type and mutant strains were found to germinate normally and grow on the leaf surface. In the wild-type, some hyphae invading the tissue through the stomata were observed at 12 hpi, and more hyphae invading directly or through the stomata occurred at 48 hpi (Figure 4b). Compared with the wild-type, the hyphae of the mutant grew more slowly, moreover, the penetration of hyphae into the leaf tissue is rare even at 48 hpi (Figure 4b). Microscopic observations of hyphal growth and dispersal in leaves at 48 hpi by wild-type mycelia, mutant mycelia or mutant with exogenous TeA are shown in Figure 4c. Hyphal dispersal and density in leaves of the *ΔHP001* mutant were decreased compared with the wild-type. Many hyphae of the wild-type already extended into the circular zone with a radius of 4 mm far from the primary inoculation site, but only a few hyphae of the mutant could extend outward into the region with a radius of 3 mm from the primary inoculation site. This indicated that an *A. alternata* mutant deficient in TeA was impaired in its ability to colonize leaves efficiently. Exogenous TeA resulted in a marked increase in hyphal expansion of the mutant, which was accompanied by an increase in hyphal distribution density (Figure 4c). In general, fungi produce toxins to induce cell death for successful invasion and disease initiation [2]. Cell death occurring in the early stages of a necrotrophic fungal infection is an indicator of successful infection. Furthermore, activation of cell death can promote colonization by necrotrophic pathogens [38]. Looking at the progression of cell death and the development of disease symptoms on leaves over the period of infection (Figure 1b), it was concluded that the hyphae invaded the leaf tissue after the cells died and obtained nutrients from the dead or dying cells for further spread and growth [6]. Therefore, TeA is crucial for *A. alternata* to break the epidermal barrier and promote the expansion and growth of invasive hyphae on the host plant.

### 3.3. TeA-Triggered ROS Production Is an Early Event in A. adenophora Infection by A. alternata

Necrotrophic pathogens can deploy diverse virulence factors that determine the disease extent to promote cell death, which is modulated by host plant hormones and ROS [6]. ROS act as signaling molecules to regulate resistance response early during infection or as a virulence factor to facilitate cell death and disease development later during infection of necrotrophic pathogens [6,39]. Our prior studies demonstrated that TeA as a novel photosynthetic inhibitor can induce chloroplast-derived ROS burst and result in cell death and leaf necrosis [19,22]. Moreover, the generation of TeA-triggered ROS in *A. adenophora* leaves is an earlier event than cell death and tissue lesion [22]. To shed light on the relationship between TeA secretion, ROS generation, and disease development during *A. alternata* infection, the accumulation of ROS in *A. adenophora* leaves after inoculation by wild-type and mutant *ΔHP001* mycelia were measured (Figure 5a,b). The generation of O_2_^•−^ was detected using dark-blue insoluble formazen as the NBT-O_2_^•−^ reaction product [40]. As shown in Figure 5a, no visible O_2_^•−^ accumulation was observed in leaves at 3, 6, and 12 h post inoculation with mutant mycelia. Inoculation of wild-type mycelia for 6 h led to a slight accumulation of O_2_^•−^ in the leaves, further being supported by a microscopic observation. At 12 hpi, much yield of O_2_^•−^ was observed in the infected area of leaves (Figure 5a). Similar results were observed when the H_2_O_2_ production in the infected leaves was assessed by the brown deposit of the DAB-H_2_O_2_ oxidized reaction product (Figure 5b) [41]. The marked accumulation of the oxidized DAB deposits was found in the infected leaves at 6 hpi by wild-type mycelia. By increasing the inoculation time, more and deeper brown deposits appeared in the infected leaves, indicating that a large amount of H_2_O_2_ is produced. Little DAB staining was observed in leaves during mutant mycelia infection (Figure 5b). Clearly, mutant *ΔHP001* deficient in TeA failed to induce H_2_O_2_ and O_2_^•−^ generation in the infected leaves. This means that TeA is absolutely necessary for *A. alternata* pathogens to trigger ROS burst in the infected plant. Since the occurrence of visible cell death is at 12 h after wild-type mycelia of *A. alternata* inoculation (Figure 1b), it is also supported that TeA-induced ROS production is an earlier event than cell death and leaf lesion during *A. alternata* infection.

To investigate the role of ROS during *A. alternata* infection, ROS scavenger CAT was applied to monitor the changes of hyphal colonization and extension, tissue damage, and disease development after leaves were inoculated with wild-type mycelia. CAT is one of the most efficient antioxidase enzymes, which can specifically decompose H_2_O_2_ molecules in cells [42]. Pretreatment with CAT significantly suppressed leaf necrotic lesion (Figure 5e,f) and photosynthetic tissue damage (Figure 5c,d) caused by wild-type mycelia. The chlorophyll fluorescence imaging as a sensitive tool can be used to assess the early damage of pathogens on leaves before the appearance of a visible disease symptom. A decrease in F_V_/F_M_ is a sensitive marker for changes in photosynthetic performance during degradation of the photosynthetic apparatus caused by pathogen infection [43]. In Figure 5e, wild-type mycelia made the color-coded images of F_O_, F_M_, and F_V_/F_M_ fade distinctly relative to the water control at 24 hpi. Pretreatment with CAT nearly eliminated these damages caused by wild-type mycelia. This is further supported by the values of fluorescence parameters F_O_, F_M_, and F_V_/F_M_ (Figure 5f). The results of early leaf damage based on fluorescence imaging are well in agreement with that of leaf necrotic lesion. This suggests that the occurrence of photosynthetic tissue destruction and leaf lesion is a result of ROS-caused oxidative damage. In fact, it had been proved that chloroplast damage is an earlier event during TeA-induced cell death [22]. These results demonstrate a strong correlation between the generation of ROS and cell death during *A. alternata* infection.

We also investigated the influence of reactive oxygen species (ROS) on the hyphal penetration, growth, and extension after inoculation with wild-type mycelia. Data from Figure 6a showed that the CAT pretreatment did not impact the germination of mycelium pieces on the surface of leaves in the early-stage of wild-type mycelia infection, but decreased evidently the frequency of hyphal penetration into leaf tissues. A comparison with wild-type mycelia without the CAT pretreatment, a lower hyphal density, and a smaller extent of hyphal distribution on leaves were observed after wild-type mycelia inoculation with the CAT pretreatment (Figure 6b). Especially, almost no hyphae on the circle leaf region with a radius of 4 mm far from the primary inoculation site was found after wild-type mycelia infection in the presence of ROS scavenger CAT (Figure 6b). Such results revealed that ROS participate in hyphal penetration, invasive hyphal growth, and extension into leaves during wild-type mycelia infection. Von Tiedemann [44] showed that ROS could assist the colonization of bean leaf tissues by necrotrophic *Botrytis cinerea* pathogens.

Based on the above analysis, it is concluded that cell death is a result of oxidative damage due to the TeA-induced ROS burst in leaves during necrotrophic *A. alternata* infection. Cell death appears to precede the successful penetration of hyphae and leaf lesion formation.

### 3.4. Endogenous AAL-Toxin and Bentazone Facilitated A. adenophora Infection by A. alternata Mutants

Although it is widely believed that cell death promotes plant disease and is an indicator of successful infection by necrotrophic pathogens, it is unclear whether this extends to all necrotrophic pathogens [6]. To further verify the role of cell death during the infection of *A. adenophora* leaves by *A. alternata*, mycotoxin AAL-toxin and herbicide bentazone that can both cause similar cell death and necrosis disease in plants were substituted for TeA. The AAL-toxin produced by *A. alternata* f. sp. *lycopersici* is a new structurally analogue of sphinganine, which causes cell death and necrosis disease in its natural host tomato and other selective plants since it stops sphingolipid biosynthesis by inhibiting ceramide synthase [11,45,46]. Bentazone is a photosynthetic inhibitor that interrupts PSII electron transport by binding to the same target D1 protein as TeA, resulting in ROS burst and cell death [19,47,48]. At 36 h post inoculation, wild-type hyphae caused evident damage to the photosynthetic activity of the primary inoculation zone where the color code of F_O_, F_M_, and F_V_/F_M_ fluorescence images almost totally became black (Figure 7a). This means that the F_V_/F_M_ value of wild-type hyphae infected leaf tissue decreased close into zero (Figure 7b). No visible change in the fluorescence images of the infected zone was found after leaves were inoculated with sterile water, AAL-toxin, bentazone or mutant hyphae separately (Figure 7a,b). However, mutant hyphae combined with exogenous AAL-toxin or bentazone significantly decreased the photosynthetic activity of the infected leaf zone (Figure 7a,b). It is indicated that AAL-toxin and bentazone can restore the ability of mutant *ΔHP001* to destroy photosynthetic tissues. As shown in Figure 7c, the loss of mutant *ΔHP001* pathogenicity was also reversed by exogenous AAL-toxin or bentazone as TeA did (Figure 1b). After inoculation for 48 h by mutant mycelia with AAL-toxin and bentazone, the infected leaves exhibited clear cell death and necrotic lesion (Figure 7c), which may be attributed to a distinct increase in the frequency of hyphal penetration and more aggressive spread of invasive hyphae.

Compared to only mutant mycelia inoculation (Figure 4b), there were really more mutant hyphae that penetrated into leaf tissues via the stomata after inoculation of mutant mycelia combined with AAL-toxin or bentazone (Figure 8a). In addition, exogenous AAL-toxin and bentazone promoted invasive hyphae to spread farther out the primary inoculation site on leaves than the mutant alone (Figure 4c and Figure 8b). These results are similar to that of TeA (Figure 4b,c). In the case of conidia inoculation, the application of endogenous TeA, AAL-toxin, and bentazone did not affect conidia germination and hyphal development at the early-stage infection (Figure 9a), but significantly enhanced hyphal entry into the leaf tissues through the stomata (Figure 9b). These results suggest that AAL-toxin and bentazone are certainly sufficient to recover the virulence of mutant *ΔHP001* pathogens. Thus, the activation and development of cell death are required for hyphal successful penetration, invasive hyphal growth, and extension, as well as disease extent in necrotrophic *A. alternata* pathogens.

## 4. Conclusions

As shown in Figure 10, a model of host disease triggered by necrotrophic *A. alternata* has been proposed on the basis of the previous studies (Chen et al., 2007, 2010) and the present evidence. When *A. alternata* pathogens directly contact the leaf surface, the toxin TeA secreted by the pathogen penetrates the mesophyll cell and inhibits photosynthetic electron transport by binding to the D1 protein, resulting in ROS burst and cell death. Subsequently, hyphae invade the leaf tissue through destroyed epidermal cells or stomata and colonize the dead photosynthetic tissue. Invasive hyphae, which extract food from the dead tissue, grow rapidly and secrete more TeA to kill the cells. The expanding hyphae invade the tissue more aggressively and invade the living tissue. Eventually, the invasive hyphae displace large portions of leaf tissue, leading to the formation of reticulate lesions as a visible symptom of disease. This study has improved our knowledge on how necrotrophic *A. alternata* pathogens infect plants and promote disease development by secreting the toxin TeA to induce cell death. However, further research is needed to elucidate the molecular mechanism of TeA-triggered disease susceptibility during infection of *A. adenophora* pathotype *A. alternata*.

## Figures and Tables

**Figure 1 cells-10-01010-f001:**
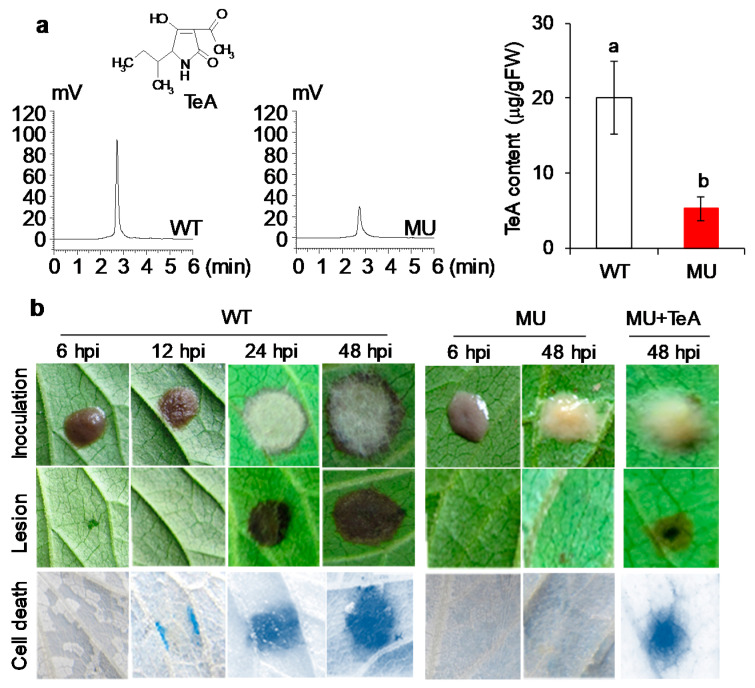
TeA is required for the virulence of *A. alternata* in *A. adenophora*. (**a**) TeA content of *A. alternata* wild-type strain (NEW001, WT) and mutant strain (*ΔHP001*, MU). The level of TeA of fresh deposited mycelia of *A. alternata* WT and MU were measured using HPLC. The results represent the means of three independent biological replicates. Data shown are mean values ± SD. Different small letters above the error bar indicate significance at 0.05 level. (**b**) Formation of cell death and leaf lesion in *A. adenophora* leaves induced by *A. alternata* WT, MU or MU with TeA. A 20 μL of WT or MU mycelial homogenates or MU mycelial homogenates with 20 μg mL^−1^ TeA (10 μL) (MU + TeA) were inoculated on the detached leaves of *A. adenophora* and incubated at 25 °C for the indicated times. Then, photographs of *A. alternata* colony (inoculation), leaf lesion, and cell death were taken. Trypan blue staining revealed leaf cell death responses. Experiments were repeated at least three times with similar results.

**Figure 2 cells-10-01010-f002:**
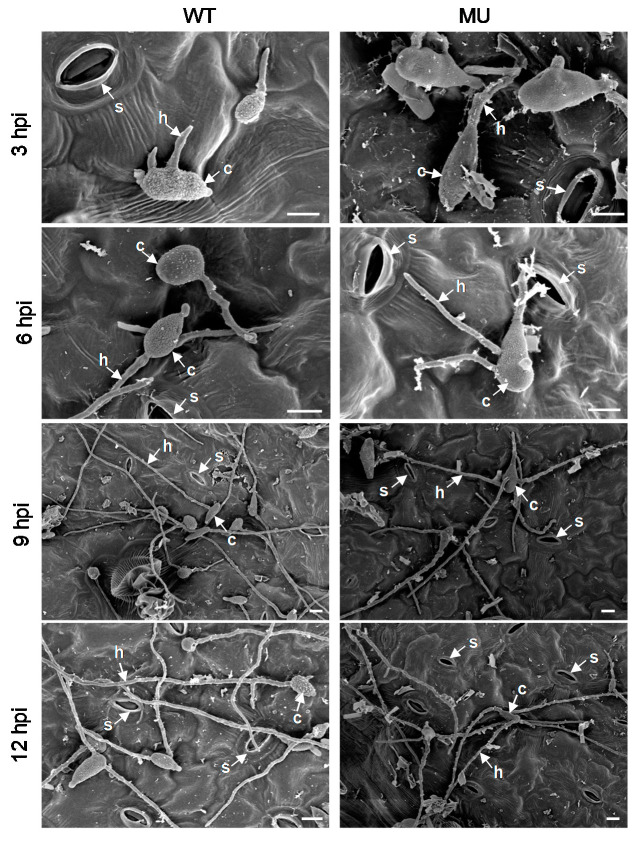
Comparison of germination and development of *A. alternata* wild-type (WT, left) and mutant *ΔHP001* (MU, right) conidia (8 × 10^5^ conidia mL^−1^) on the surface of *A. adenophora* leaves. Scanning electron micrographs were shown at 3, 6, 9, and 12 h post inoculation (hpi). Note that no differences between MT and MU were present during germination of conidia and growth of hyphae at the early-stage infection. Here, s: Stomata; c: Conidium; h: Hypha. Scale bars = 10 μm. One representative result out of three repetitions is presented.

**Figure 3 cells-10-01010-f003:**
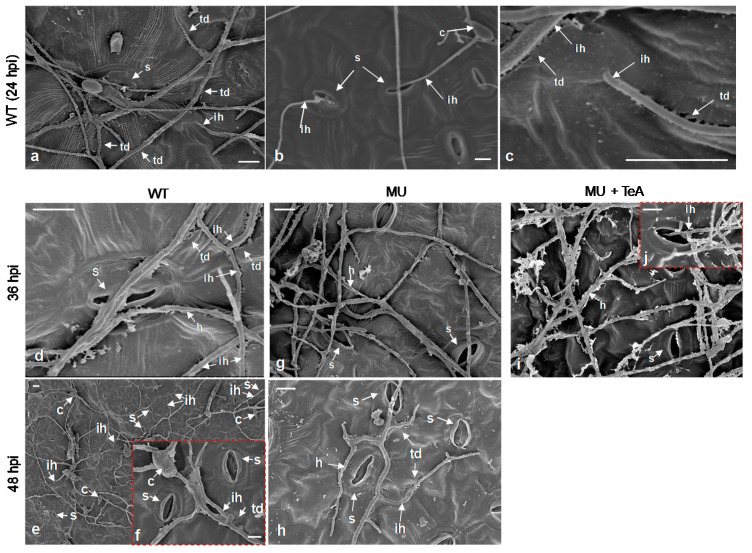
TeA is important for hyphal penetration of *A. adenophora* by *A. alternata*. *A. adenophora* leaves inoculated with *A. alternata* wild-type (WT, **a**–**f**), mutant *ΔHP001* (MU, **g**,**h**), conidia (8 × 10^5^ conidia mL^−1^) or mutant conidia with TeA (20 μg mL^−1^, MU+TeA, **i**,**j**) were observed using scanning electron microscopy at 24, 36, and 48 h post inoculation (hpi). Note that visual differences between WT and MU were present during hyphal penetration and invasive hyphal colonization. Here, s: Stomata; c: Conidium; h: Hypha; ih: Invasive hypha; td: Tissue destroyed. Scale bars = 10 μm. One representative result out of three repetitions is presented.

**Figure 4 cells-10-01010-f004:**
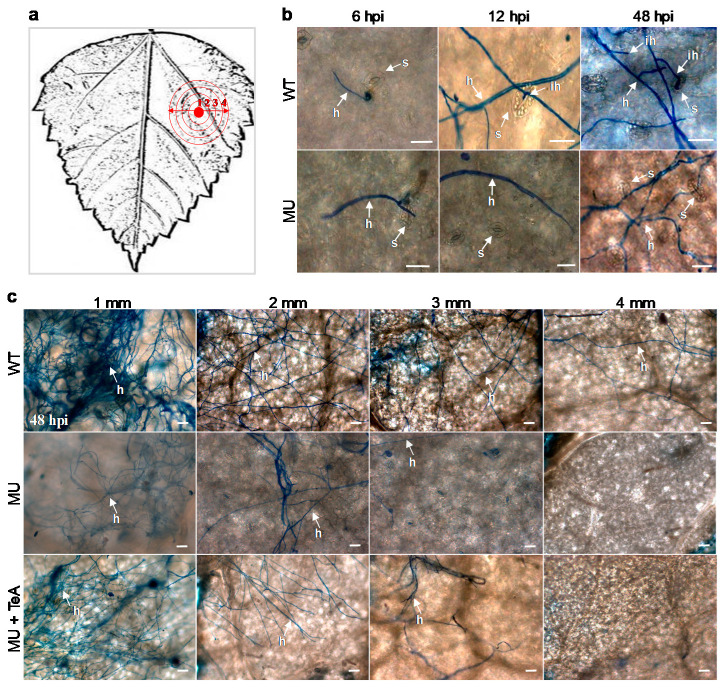
TeA is important for invasive hyphal growth and development of *A. alternata*. *A. adenophora* leaves were inoculated by *A. alternata* wild-type (20 μL, WT), mutant *ΔHP001* (20 μL, MU) mycelial homogenates or MU mycelial homogenates (20 μL) with 20 μg mL^−1^ TeA (10 μL) (MU + TeA) and incubated at 25 °C for the indicated times. (**a**) Experimental model of mycelial inoculation and sample collection. The red closed circle in the center indicates the direct-contact site of mycelial inoculation on the surface of leaves; the red open circles indicate the demarcations of an area around the primary inoculation site. Here, 1, 2, 3, and 4 represent the circle with a radius of 1, 2, 3, and 4 mm from the primary inoculation site outwards, respectively. (**b**) Hyphal growth and penetration during the early-stage of *A. adenophora* infection by *A. alternata* WT or MU mycelial homogenates. Shown are visual differences in hyphal penetration between *A. alternata* WT and MU. (**c**) The effect of TeA on invasive hyphal growth and distribution of *A. alternata* on *A. adenophora* leaves. Shown are visible differences in invasive hyphal growth and extension between *A. alternata* WT, MU, and MU with TeA. Tissues were stained with lactophenol-trypan blue to visualize hyphae. Here, s: Stomata; h: Hypha; ih: Invasive hypha. Scale bars = 20 μm. Experiments were repeated at least three times with similar results.

**Figure 5 cells-10-01010-f005:**
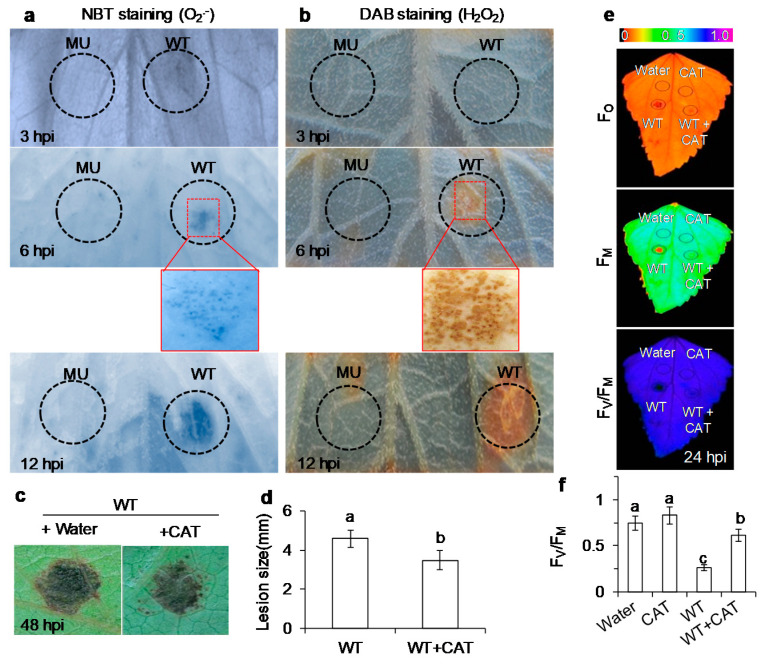
TeA-induced ROS burst is involved in *A. adenophora* infection by *A. alternata*. (**a**,**b**) The accumulation of ROS in *A. adenophora* leaves inoculated with *A. alternata* wild-type (20 μL, WT) and mutant *ΔHP001* (20 μL, MU) mycelial homogenates for 3, 6, and 12 h. NBT and DAB staining were used to visualize the production of superoxide anion (O_2_^•−^) and hydrogen peroxide (H_2_O_2_). Microscopic observations were made at a 10× magnification. Experiments were repeated at least three times with similar results. The ROS scavenger catalase reduced *A. alternata*-caused necrotic lesions (**c**) and their average diameter (**d**) and photosynthetic damage (**e**,**f**) on *A. adenophora* leaves. *A. adenophora* leaves pretreated with 300 U mL^−1^ catalase or sterile water were inoculated with *A. alternata* wild-type mycelial homogenates (20 μL) or without *A. alternata* for 24 or 48 h, and then chlorophyll fluorescence imaging and leaf lesions were measured. Fluorescence images were indicated by a color code in the order of black (0) through red, orange, yellow, green, blue, violet to purple (1). Images of F_O_ (minimal fluorescence), F_M_ (maximal fluorescence), and F_V_/F_M_ (the maximum quantum yield of PSII) (**e**) and the values of F_V_/F_M_ (**f**) were shown. Black open circles on leaf images indicate the primary inoculation sites. Data shown are mean values ± SD of three independent biological replicates. The different small letters above the error bar mean a significant difference at 0.05 level. CAT: Catalase.

**Figure 6 cells-10-01010-f006:**
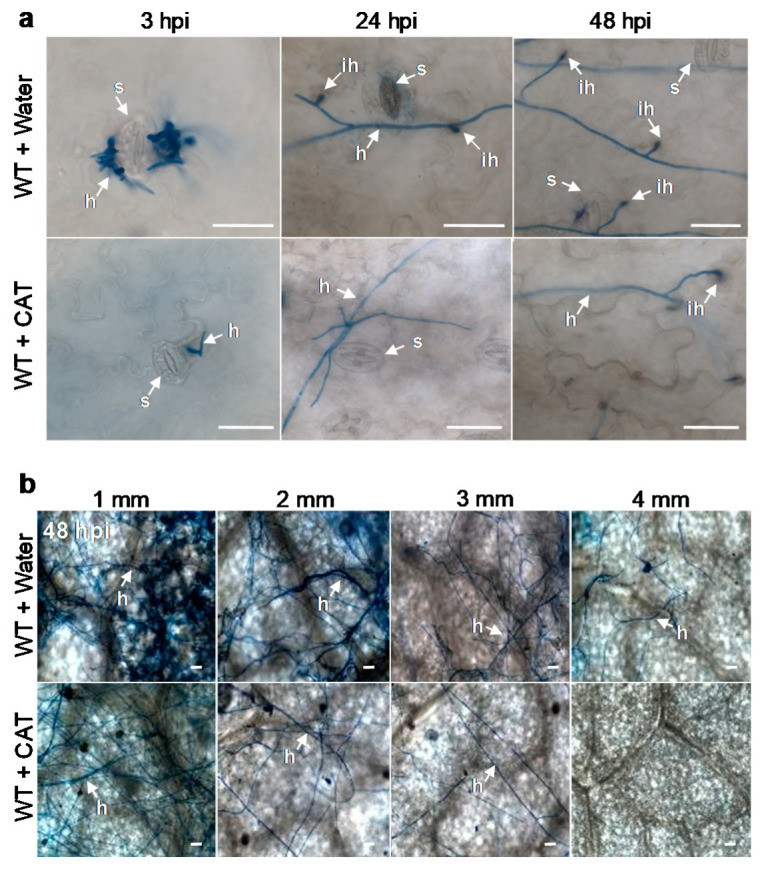
ROS scavenger catalase suppressed hyphal penetration and invasive hyphal growth of *A. alternata*. *A. adenophora* leaves pretreated with 300 U mL^−1^ catalase or sterile water were inoculated with *A. alternata* wild-type mycelial homogenates (20 μL) for the indicated times. (**a**) The effect of catalase on hyphal growth and penetration during the early-stage at 3, 24, and 48 h post inoculation (hpi). (**b**) The effect of catalase on invasive hyphal growth and extension on *A. adenophora* leaves inoculated by *A. alternata* at 48 h post inoculation (hpi). Tissues were stained with lactophenol-trypan blue to visualize hyphae. Here, s: Stomata; h: Hypha; ih: Invasive hypha; CAT: Catalase. Scale bars = 20 μm. Experiments were repeated at least three times with similar results.

**Figure 7 cells-10-01010-f007:**
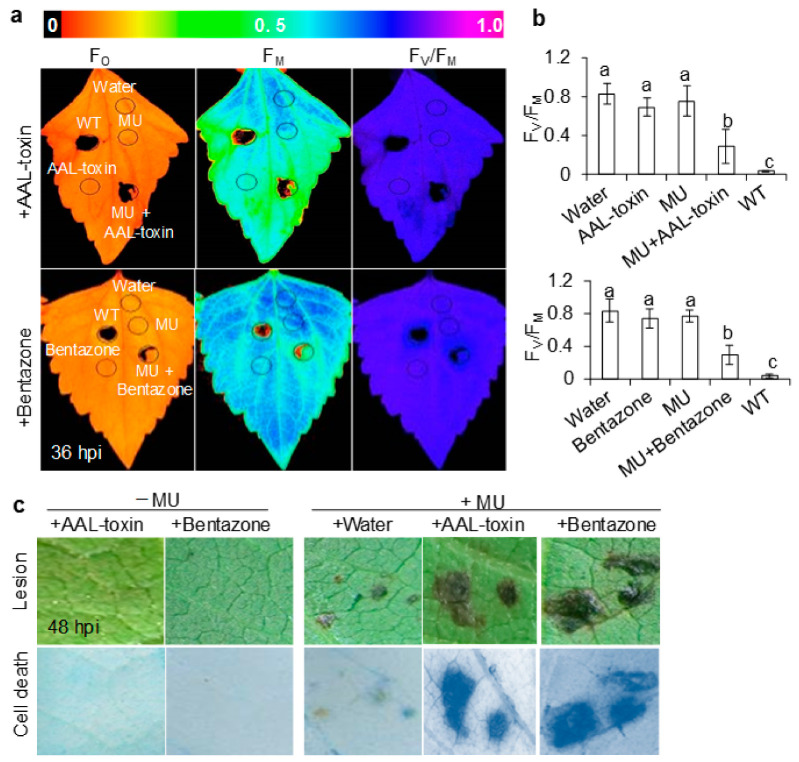
Endogenous AAL-toxin and bentazone increased susceptibility to *A. alternata* mutant *ΔHP001*. *A. adenophora* leaves were inoculated by sterile water (10 μL), 20 μg mL^−1^ AAL-toxin (10 μL), 200 μg mL^−1^ bentazone (10 μL), *A. alternata* wild-type mycelial homogenates (20 μL, WT), mutant *ΔHP001* mycelial homogenates (20 μL, MU), mutant *ΔHP001* mycelial homogenates (20 μL) with 20 μg mL^−1^ AAL-toxin (10 μL) (MU + AAL-toxin) or 200 μg mL^−1^ bentazone (10 μL) (MU + bentazone). (**a**,**b**) The effect of application of AAL-toxin and bentazone on the photosynthetic activity of *A. adenophora* leaves inoculated by *A. alternata* mutant *ΔHP001* at 36 h post inoculation (hpi). Chlorophyll fluorescence imaging of F_O_ (minimal fluorescence), F_M_ (maximal fluorescence), and F_V_/F_M_ (the maximum quantum yield of PSII) (**a**) and the values of F_V_/F_M_ (**b**) were shown. Fluorescence images were indicated by a color code in the order of black (0) through red, orange, yellow, green, blue, violet to purple (1). The black open circles on leaf images indicate the primary inoculation sites. Data shown are mean values ± SD of three independent biological replicates. Different small letters above the error bar mean a significant difference at 0.05 level. (**c**) The effect of application of AAL-toxin and bentazone on leaf necrotic lesion and cell death of *A. adenophora* inoculated by *A. alternata* mutant *ΔHP001* at 48 h post inoculation (hpi). Trypan blue staining revealed leaf cell death responses. Microscopic observations were made at a 10× magnification. Experiments were repeated three times with similar results.

**Figure 8 cells-10-01010-f008:**
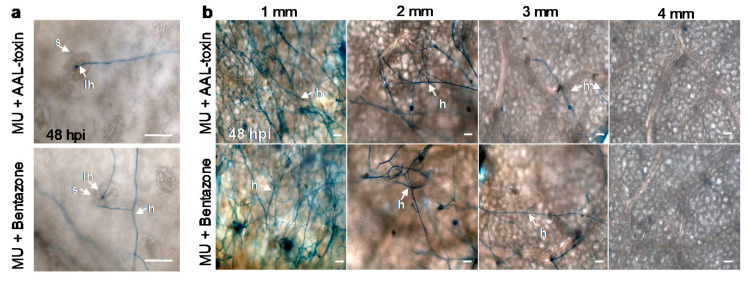
Endogenous AAL-toxin and bentazone increased hyphal penetration and invasive hyphal growth of *A. alternata* mutant *ΔHP001*. *A. adenophora* leaves were inoculated by mutant *ΔHP001* mycelial homogenates (20 μL, MU), mutant *ΔHP001* mycelial homogenates (20 μL) with 20 μg mL^−1^ AAL-toxin (10 μL) (MU + AAL-toxin) or 200 μg mL^−1^ bentazone (10 μL) (MU + bentazone). (**a**) The effect of endogenous AAL-toxin and bentazone on hyphal penetration on *A. adenophora* leaves inoculated by *A. alternata* at 48 hpi. (**b**) The effect of endogenous AAL-toxin and bentazone on invasive hyphal growth and extension on *A. adenophora* leaves inoculated by *A. alternata* at 48 hpi. Lactophenol-trypan blue was used to visualize hyphae. Here, s: Stomata; h: Hypha; ih: Invasive hypha. Scale bars = 20 μm. Experiments were repeated three times with similar results.

**Figure 9 cells-10-01010-f009:**
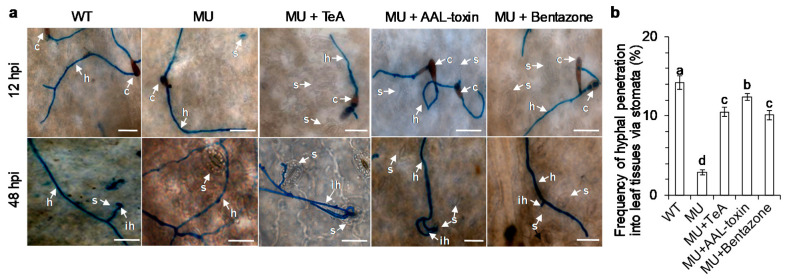
Effect of endogenous TeA, AAL-toxin, and bentazone on conidia germination, hyphal growth, and penetration on *A. adenophora* inoculated by *A. alternata* mutant *ΔHP001*. *A. adenophora* leaves were inoculated by *A. alternata* wild-type (WT) conidia (8 × 10^5^ conidia mL^−1^), mutant *ΔHP001* (MU) conidia (8 × 10^5^ conidia mL^−1^), MU conidia (8 × 10^5^ conidia mL^−1^) with 20 μg mL^−1^ TeA (10 μL) (MU + TeA) or 20 μg mL^−1^ AAL-toxin (10 μL) (MU + AAL-toxin) or 200 μg mL^−1^ bentazone (10 μL) (MU + bentazone). (**a**) Conidia germination and development and hyphal penetration at 12 and 48 hpi. Here, s: Stomata; c: Conidium; h: Hypha; ih: Invasive hypha. Scale bars = 20 μm. (**b**) The frequency of hyphal penetration into leaf tissues via the stomata. The tissue was stained with lactophenol-trypan blue to visualize hyphae. Hypha from at least 300 stomata were counted. Data shown are mean values ± SD of three independent biological replicates. Different small letters above the error bar mean a significant difference at 0.05 level.

**Figure 10 cells-10-01010-f010:**
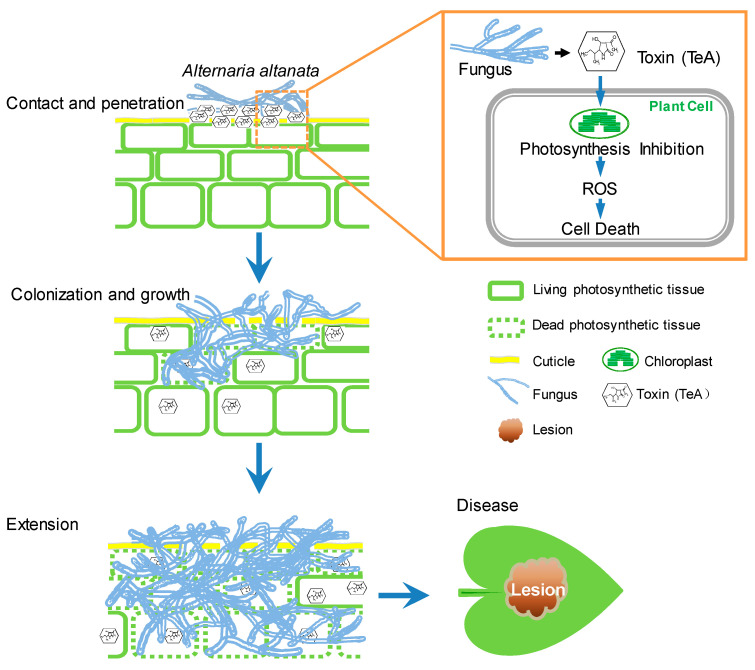
Model for a role of TeA in *A. alternata* infection. *A. alternata* grows subcuticularly and destroys epidermal cells by secreting mycotoxin TeA as its major virulence factor. TeA inhibits photosynthetic activity, which induces ROS burst and mesophyll cell death. Nutrition from the dead cell further promotes growth and extension of invasive hyphae in leaf tissues. Eventually, *A. alternata* hyphae replace large parts of leaf tissues, leading to necrotic lesion formation as a visible disease symptom. Here, hyphae are shown in light blue; photosynthetic tissue in green; plant cuticle in yellow; necrotic lesion tissue in brown; solid green lines indicate the living tissue; dashed green lines indicate the dead tissue.

## Data Availability

All data presented in this study is contained in main text.

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
