# Peer review of "Tenuazonic Acid-Triggered Cell Death Is the Essential Prerequisite for Alternaria alternata (Fr.) Keissler to Infect Successfully Host Ageratina adenophora"

_cells, 2021, doi:10.3390/cells10051010_

Round 1

Reviewer 1 Report

The study is well conducted. The article is clear and well structured. However, I’ve a few comments:

1 -line 110: please define the ∆HP001 mutation. Which gene is affected ? What is the product of the gene ?

2 -the conclusion contained in lines 493-494 is not so much supported by figure 9. In this figure, it is difficult to see that the infection is more important at the stomata loci.

Conclusion:

3 -I don’t understand if the statement lines 521-522 is supported by the present work. If so, it is over-interpretated. If the conclusion comes from literature, references should be added.

Author Response

Point 1: line 110: please define the ∆HP001 mutation. Which gene is affected? What is the product of the gene?

RE: Thanks for your hard work and kind suggestion. The HP001 gene was knocked out in mutant ∆HP001. The HP001 protein affects the yield of TeA by the Fus3/Kss1 MAPK signaling pathway in Alternaria alternata. The additional information was added to describe the ∆HP001 mutant (MU, HP001 gene knockout strain) in the revision. However, the research on the function and regulation network of HP001 gene did not being published, so we could not give more its details in this paper.

 Point 2: -the conclusion contained in lines 493-494 is not so much supported by figure 9. In this figure, it is difficult to see that the infection is more important at the stomata loci.

RE: Thanks for your hard work. The conclusion “application of endogenous TeA, AAL-toxin and bentazone significantly enhanced hyphal entry into the leaf tissues through stomata” can be supported by Figure 9b not Figure 9a (we deleted the figure 9a in the revision). Based on at least 300 stomata, the frequency of hyphal penetration into leaf tissues via stomata were counted in Figure 9b. The result is clear.

Point 3: Conclusion: 3 -I don’t understand if the statement lines 521-522 is supported by the present work. If so, it is over-interpretated. If the conclusion comes from literature, references should be added.

RE: Thanks for your good suggestion. The mode is proposed on the basis of our previous studies and the present work. So we modified the statement according to your suggestion, and added the references.

Reviewer 2 Report

The manuscript by Shi et al. investigated the influence of tenuazonic acid on cell death. In general, it is an interesting topic and worth publishing. However, prior to publication, the manuscript have to be revised with regard to certain inconsistencies and some things have to be clarified. Please find below my comments, including both minor (style) suggestions and major comments which have to be solved. Title: In my opinion it is not a good idea to use the abbreviated form of tenuazonic acid. I would recommend to write the fill name. Line 15: I think that the sentence should be as follows “Here, an A. alternata wild-type and the toxin-deficient mutant deltaHP001 with a 75% reduction in TeA production were used. Line 18: Please explain the abbreviation of CAT since it is used here for the first time. Line 43: What do you mean with “successful”? I am not sure whether it is the correct expression in this context. General remark for the materials and methods section: In general, the following should be stated for all the materials (company, city, if US state/country). Please consider this throughout the whole manuscript since in most cases you only state the company. Line 100: It should be “healthy” instead of “health”. Line 115: Please explain the abbreviation “PDB” Line 117, 144: Please provide the centrifugation speed in g-force (or relative centrifugation force, rcf) instead of rounds per minutes (rpm). When you state the later one you should specify the type of centrifuge and rotor. Line 131: How did you identify the pretreated site, did you mark it somehow? The applied volume is quite low (1 µL) an most probably not visible. Line 133: Why did you use 5 µL instead of 1 µL and why did you use so low volumes? Line 146: In the future please consider alternatives for chloroform to apply more “green solutions” and avoid the use of halogenated solvents. Line 154: Could you please specify the material of the filter. Depending on the filter material, some compounds might get trapped on the filter. Thanks! Line 156: The second time, the “t” at the end of “Agilent” is missing. Line 159: You report here that the retention time of TeA was 10 min, but according to Figure 1A, the retention time seems more to be around 3 min. Why is there such a discrepancy? Did you apply different methods? Please go back to the raw data and check it. The applied column is rather long for just a single analyte detection. Line 165: The dot after “et al.” is missing Line 170: It should be “at room temperature” instead of “in room temperature”. Figure 2: Why did you not use the same picture frame for all pictures? At least according to the location of the stomata it is not same. Furthermore. Could you please increase the information of the scale bar – it is hardly visible and you have to zoom in quite a lot. Furthermore, please mention that different scale bars are used. It would have been good if also for WT 12 hpi 100 µm would have been used to be consistent with the other 9 hpi and 12 hpi pictures. To include the day of the picture is not advantageous in your case since it seems to be quite old picture (2012 or 2014 depending on the style of display. Figure 3: For this figure more information is needed in the Figure caption. Why did you sometimes zoom in and why is there an overlap of some sub-figures. Furthermore, the same statements with regard to the scale bars is true as in Figure 2. Why is there no special letter assigned to the lay in in subfigure J? On top of that in the figure itself you report 24, 36 and 48 hpi, but in the caption 24, 36 and 96 hpi. Please check and correct. Figure 4, 5, 6, 7, 8 and 9: Could you please add a scale bars.

Author Response

Reviewer 2

The manuscript by Shi et al. investigated the influence of tenuazonic acid on cell death. In general, it is an interesting topic and worth publishing. However, prior to publication, the manuscript has to be revised with regard to certain inconsistencies and some things have to be clarified. Please find below my comments, including both minor (style) suggestions and major comments which have to be solved.

RE: Thanks for your hard work and kind comments.

Point 1: Title: In my opinion it is not a good idea to use the abbreviated form of tenuazonic acid. I would recommend to write the fill name.

RE: The fill name was used in the revision.

Point 2: Line 15: I think that the sentence should be as follows “Here, an A. alternata wild-type and the toxin-deficient mutant deltaHP001 with a 75% reduction in TeA production were used.

RE: Thanks for your good suggestion. We agreed with you and modified it in the revision.

Point 3: Line 18: Please explain the abbreviation of CAT since it is used here for the first time.

RE: Thanks. We used catalase instead of CAT here.

 Point 4: Line 43: What do you mean with “successful”? I am not sure whether it is the correct expression in this context.

RE: “Successful” means Alternaria is a great success in fungus families. Because Alternaria species have different lifestyles, a wide variety of host ranges and worldwide distribution, diverse phytotoxins. So, Alternaria is a highly successful group in fungus families.

Point 5: General remark for the materials and methods section: In general, the following should be stated for all the materials (company, city, if US state/country). Please consider this throughout the whole manuscript since in most cases you only state the company.

RE: Thanks for your good suggestion. We added the information of city and country in the materials section.

Point 6: Line 100: It should be “healthy” instead of “health”.

RE: We are sorry for this mistake. We modified in the revision.

Point 7: Line 115: Please explain the abbreviation “PDB”

RE: We added the explanation. PDB means “potato dextrose broth”.

Point 8: Line 117, 144: Please provide the centrifugation speed in g-force (or relative centrifugation force, rcf) instead of rounds per minutes (rpm). When you state the later one you should specify the type of centrifuge and rotor.

RE: Thanks for your good advice. We modified the part in the revision. The 1 000 g is instead of 3 000 rpm.

Point 9: Line 131: How did you identify the pretreated site, did you mark it somehow? The applied volume is quite low (1 µL) an most probably not visible.

RE: We are sorry for this mistake. After we checked the original document, it was found that 10 µL of 20 μg/mL TeA (not 200 μg/mL), 20 μg/mL AAL-toxin and 200 μg/mL betazone were used in our experiments. We corrected this mistake in the revision.

Before the liquid drop was added onto the surface of leaves, the site was selected and recorded using a camera. Furthermore, a visible sign on the leaf surface could be observed after the drop of TeA, or AAL-toxin or bentazone dried.  

Point 10: Line 133: Why did you use 5 µL instead of 1 µL and why did you use so low volumes?

RE: When the experiments of catalase pretreatment were carried out, we found that 5 and 10 µL of 300 U mL-1 CAT showed clear effect. Finally, 5 µL not 10 µL was selected in order to save the CAT solution. 

Point 11: Line 146: In the future please consider alternatives for chloroform to apply more “green solutions” and avoid the use of halogenated solvents.

RE: Thanks for your kind suggestion. It’s very important to use “green solution” in all experiments.

Point 12: Line 154: Could you please specify the material of the filter. Depending on the filter material, some compounds might get trapped on the filter. Thanks!

RE: You are right. In our experiments, the Whatman GF/B 0.22 μm glass microfiber filter was used. We added the detail information in the revision.

Point 13: Line 156: The second time, the “t” at the end of “Agilent” is missing.

RE: We are sorry for such mistake. It was corrected as “Agilent”.

Point 14: Line 159: You report here that the retention time of TeA was 10 min, but according to Figure 1A, the retention time seems more to be around 3 min. Why is there such a discrepancy? Did you apply different methods? Please go back to the raw data and check it. The applied column is rather long for just a single analyte detection.

RE: We are sorry for the wrong understanding about the retention time. In our experiments, the whole running time is 10 min, which was incorrectly regarded as the retention time. The retention time of TeA was 2.73 min. Because we found a mistake in Figure 1A, new Figure 1 was used in the revision.

As you mention, the applied column was run for too long to check TeA. In the future research, we will short the time into 4 min.

Point 15: Line 165: The dot after “et al.” is missing

RE: Thanks. We added the dot after Joo et al.

Point 16: Line 170: It should be “at room temperature” instead of “in room temperature”.

RE: Thanks. The mistake was corrected in the revision.

Point 17: Figure 2: Why did you not use the same picture frame for all pictures? At least according to the location of the stomata it is not same. Furthermore. Could you please increase the information of the scale bar – it is hardly visible and you have to zoom in quite a lot. Furthermore, please mention that different scale bars are used. It would have been good if also for WT 12 hpi 100 µm would have been used to be consistent with the other 9 hpi and 12 hpi pictures. To include the day of the picture is not advantageous in your case since it seems to be quite old picture (2012 or 2014 depending on the style of display.

RE: Thanks for your hard work and good suggestions. We rearranged each photo in Figure 2, and added the scale bars and moved the date information. So, new Figure 2 was used in the revision.

Point 18: Figure 3: For this figure more information is needed in the Figure caption. Why did you sometimes zoom in and why is there an overlap of some sub-figures. Furthermore, the same statements with regard to the scale bars is true as in Figure 2. Why is there no special letter assigned to the lay in in subfigure J? On top of that in the figure itself you report 24, 36 and 48 hpi, but in the caption 24, 36 and 96 hpi. Please check and correct.

RE: Thanks. As you said, there is some confusion in the arrangement of figures. We rearranged and corrected them according to your suggestions.

First, an enlarged view of Figure H (Figure I) was deleted because the results are enough clear in Figure H. So, the old Figure J and the subfigure were marked as “I” for the major figure and “J” for the insert figure.

The scale bars were added in each figures without the date information. So, new Figure 3 was used in the revision.

The mistake in the caption was corrected. The time should be 24, 36 and 48 h.

Point 19: Figure 4, 5, 6, 7, 8 and 9: Could you please add a scale bars.

RE: Thanks. The scale bars were added in Figures 4, 6, 8-9. So, new Figure 4, 6, 8-9 were used in the revision. Results of Figure 5 and Figure 7 were observed using a stereo microscope without a scale. So, no scale bars were added in this two figures.

Reviewer 3 Report

The manuscript describes an original research and brings important new information that is highly valuable to the field. However, there are some areas that can be improved, which authors can find in the text below.

There are minor spelling and grammar mistakes present in the manuscript, therefore overall English and typography check is advised.  Some general errors in need of revision include singular/plural of nouns (e.g. line 91), and generic typing errors.

Introduction:

  1. Line 59: “alernaric acid” - Probably alternaric acid?
  2. Line 91: “disease development when the host A. adenophora leaves is infected“

Materials and methods:

  1. Line 99-100: Please refrain from using “about” and “approximately” as descriptors. Either the humidity was 70 % or it was not. If it was not the same throughout the experiment, use an average value - in case there was measuring done. How did you monitor/measure light quantity and humidity?
  2. Line 116: “under dark” - Either use “under dark conditions” or “in dark”.
  3. Line 139: “at 25 °C in at high humidity”
  4. Line 144: “by centrifuged”
  5. Parts of experimental design are missing. How many samples were created? How many repetitions were done and used for statistical analysis?

Results and discussion:

  1. Line 217: “three-quarters lower” - Three times lower, not three-quarters.

Author Response

The manuscript describes an original research and brings important new information that is highly valuable to the field. However, there are some areas that can be improved, which authors can find in the text below.

 There are minor spelling and grammar mistakes present in the manuscript, therefore overall English and typography check is advised.  Some general errors in need of revision include singular/plural of nouns (e.g. line 91), and generic typing errors.

 RE: Thanks for your hard work. We modified the English carefully and corrected many mistakes. Thanks for your kind advice again.

Point 1: Introduction:

Line 59: “alernaric acid” - Probably alternaric acid?

RE: We are sorry for such mistake. It is alternaric acid. We corrected it in the revision.

Point 2: Line 91: “disease development when the host A. adenophora leaves is infected”

  RE: Thanks. We corrected the mistake. It should be “…… leaves were infected”.

Point 3: Materials and methods:

Line 99-100: Please refrain from using “about” and “approximately” as descriptors. Either the humidity was 70 % or it was not. If it was not the same throughout the experiment, use an average value - in case there was measuring done. How did you monitor/measure light quantity and humidity?

 RE: Thanks for your good suggestion. Usually, a Field Scout® quantum meter (Spectrum Inc., USA) and a temperature and humidity meter (Toprie Inc., Shengzhen, China) were used to measure the light intensity and relative humidity. So, we corrected the part according to our experimental data.   

Point 4: Line 116: “under dark” - Either use “under dark conditions” or “in dark”.

 RE: We used “under dark conditions” in the revision.

Point 5: Line 139: “at 25 °C in at high humidity”

 RE: We are sorry for this mistake. We corrected as “at 25 °C and high relative humidity”.

Point 6: Line 144: “by centrifuged”

 RE: It should be “by centrifugation”. We corrected in the revision.

Point 7: Parts of experimental design are missing. How many samples were created? How many repetitions were done and used for statistical analysis?

RE: Thanks for your good suggestion. In fact, we described the detail information of experimental repetitions in the caption of some figures. Based on your suggestion, we added the detail information in the caption of each figures in the revision.

Point 8: Results and discussion:

Line 217: “three-quarters lower” - Three times lower, not three-quarters.

RE: We corrected the mistake according to your advice.

Round 2

Reviewer 2 Report

Thank you very much for the revision. I have only commented your replies in case it was necessary. Please find below my comments to the revised version.

Line 60: Please also explain the abbreviation “ROS”. Sorry, I missed it the first time.

>>> RE: “Successful” means Alternaria is a great success in fungus families. Because Alternaria species have different lifestyles, a wide variety of host ranges and worldwide distribution, diverse phytotoxins. So, Alternaria is a highly successful group in fungus families.

Reviewer’s reply: Ok, thank you very much for this explanation. I am still not convinced and would recommend to use “highly diverse”.

>>> RE: Thanks for your good suggestion. We added the information of city and country in the materials section.

Reviewer’s reply: Ok, more or less, there are still several cities missing. However, it is up to the editor whether it is acceptable or not.

>>> Point 9: Line 131: How did you identify the pretreated site, did you mark it somehow? The applied volume is quite low (1 µL) an most probably not visible.

RE: We are sorry for this mistake. After we checked the original document, it was found that 10 µL of 20 μg/mL TeA (not 200 μg/mL), 20 μg/mL AAL-toxin and 200 μg/mL betazone were used in our experiments. We corrected this mistake in the revision.

Before the liquid drop was added onto the surface of leaves, the site was selected and recorded using a camera. Furthermore, a visible sign on the leaf surface could be observed after the drop of TeA, or AAL-toxin or bentazone dried.

Reviewer’s reply: This is a major issue. How much AAL-toxin and betazone did you now apply? You have changed the volume and the concentration for TeA (the totally applied amount stays the same), but in case of AAL-toxin and betazone you did not change it. Does this mean that the concentration was 10 times higher than previously stated?

Point 14: Line 159: You report here that the retention time of TeA was 10 min, but according to Figure 1A, the retention time seems more to be around 3 min. Why is there such a discrepancy? Did you apply different methods? Please go back to the raw data and check it. The applied column is rather long for just a single analyte detection.

RE: We are sorry for the wrong understanding about the retention time. In our experiments, the whole running time is 10 min, which was incorrectly regarded as the retention time. The retention time of TeA was 2.73 min. Because we found a mistake in Figure 1A, new Figure 1 was used in the revision.

As you mention, the applied column was run for too long to check TeA. In the future research, we will short the time into 4 min.

Reviewer’s reply: Ok, now it makes more sense, but I am in doubt that you achieved a sufficient retention. The analyte should elute after two times the dead volume. To be honest, I am not sure whether this is the case in your application. The 250x4,6 mm 5 µM column is quite huge.

>>> Point 17: Figure 2: Why did you not use the same picture frame for all pictures? At least according to the location of the stomata it is not same. Furthermore. Could you please increase the information of the scale bar – it is hardly visible and you have to zoom in quite a lot. Furthermore, please mention that different scale bars are used. It would have been good if also for WT 12 hpi 100 µm would have been used to be consistent with the other 9 hpi and 12 hpi pictures. To include the day of the picture is not advantageous in your case since it seems to be quite old picture (2012 or 2014 depending on the style of display.

RE: Thanks for your hard work and good suggestions. We rearranged each photo in Figure 2, and added the scale bars and moved the date information. So, new Figure 2 was used in the revision.

Reviewer’s reply: Thank you very much. You have not answered my first questions. Why did you not use the same picture frame for all pictures of the same set-up (hence one for WT and one for MU).

RE: Thanks. The scale bars were added in Figures 4, 6, 8-9. So, new Figure 4, 6, 8-9 were used in the revision. Results of Figure 5 and Figure 7 were observed using a stereo microscope without a scale. So, no scale bars were added in this two figures.

Reviewer’s reply: Thank you very much for the modifications, it looks more professional now. In case of Figure 5 and 7, could you please state the magnifications. Thanks!
